# α-IRAK-4 Suppresses the Activation of RANK/RANKL Pathway on Macrophages Exposed to Endodontic Microorganisms

**DOI:** 10.3390/ijms25158434

**Published:** 2024-08-02

**Authors:** Elsa Montserrat Hernández-Sandoval, Raquel Sánchez-Gutiérrez, Ana Patricia Torres-Monjarás, Diana Lorena Alvarado-Hernández, Verónica Méndez-González, Berenice Hernández-Castro, Sofía Bernal-Silva, Andreu Comas-García, Ricardo Martínez-Rider, Roberto González-Amaro, Marlen Vitales-Noyola

**Affiliations:** 1Endodontics Postgraduate Program, Faculty of Dentistry, Autonomous University of San Luis Potosi, Manuel Nava 2, San Luis Potosi 78290, SLP, Mexico; 3.elsah@gmail.com (E.M.H.-S.); patiie@hotmail.com (A.P.T.-M.); veronica.mendez@uaslp.mx (V.M.-G.); 2Department of Immunology, School of Medicine, Autonomous University of San Luis Potosi, San Luis Potosi 78210, SLP, Mexico; r.sanchez@ttuhsc.edu (R.S.-G.); diana.alvarado@uaslp.mx (D.L.A.-H.); berenice.castro@uaslp.mx (B.H.-C.); rgonzale@uaslp.mx (R.G.-A.); 3Department of Molecular and Translational Medicine, School of Medicine, Texas Tech University Health Sciences, El Paso, TX 79905, USA; 4Department of Microbiology, Faculty of Medicine, Autonomous University of San Luis Potosi, San Luis Potosi 78210, SLP, Mexico; sofia.bernal@uaslp.mx (S.B.-S.); andreu.comas@uaslp.mx (A.C.-G.); 5School of Medicine, Cuauhtemoc University, Manuel Nava 3291, San Luis Potosi 78290, SLP, Mexico; 6Oral and Maxillofacial Surgery Specialty, Faculty of Dentistry, Autonomous University of San Luis Potosi, Manuel Nava 2, San Luis Potosi 78290, SLP, Mexico; rmrider@uaslp.mx

**Keywords:** *Enterococcus faecalis*, IRAK-4, lipoteichoic acid, macrophages, osteoclasts, RANK, RANKL

## Abstract

Periapical lesions are common pathologies affecting the alveolar bone, often initiated by intraradicular lesions resulting from microbial exposure to dental pulp. These microorganisms trigger inflammatory and immune responses. When endodontic treatment fails to eliminate the infection, periapical lesions persist, leading to bone loss. The RANK/RANKL/OPG pathway plays a crucial role in both the formation and the destruction of the bone. In this study, the objective was to inhibit the RANK/RANKL pathway in vitro within exposed Thp-1 macrophages to endodontic microorganisms, specifically *Enterococcus faecalis*, which was isolated from root canals of 20 patients with endodontic secondary/persistent infection, symptomatic and asymptomatic, and utilizing an α-IRAK-4 inhibitor, we introduced endodontic microorganisms and/or lipoteichoic acid from *Streptococcus* spp. to cellular cultures in a culture plate, containing thp-1 cells and/or PBMC from patients with apical periodontitis. Subsequently, we assessed the percentages of RANK+, RANKL+, and OPG+ cells through flow cytometry and measured the levels of several inflammatory cytokines (IL-1β, TNF-α, IL-6, IL-8, IL-10, and IL-12p70) in the cellular culture supernatant through a CBA kit and performed analysis by flow cytometry. A significant difference was observed in the percentages of RANK+RANKL+, OPG+ RANKL+ cells in thp-1 cells and PBMCs from patients with apical periodontitis. The findings revealed significant differences in the percentages of the evaluated cells, highlighting the novel role of the IRAK-4 inhibitor in addressing this oral pathology, apical periodontitis, where bone destruction is observed.

## 1. Introduction

Apical periodontitis refers to an acute and painful inflammation of the periodontal ligament, leading the destruction of periradicular tissues. This condition can arise from trauma or infection entering through root canals, resulting from the interplay between microbial factors and the host immune response [1]. Various microorganisms have been implicated in these infections, each possessing distinct virulence factors. These include *Fusobacterium* sp., *Prevotella* sp., *Porphyromonas* sp., and *Actinomyces* sp., as well as facultative anaerobic Gram-positive coccus and bacilli, such as *Streptococcus* sp., *Enterococcus* sp., and *Peptostreptococcus* sp. [2,3]. *Enterococcus faecalis* stands out as the most frequently isolated pathogen in periradicular infections, with a prevalence of 24–77% [3].

These microorganisms exhibit resilience in extreme environmental conditions and harbor various virulence factors, such as lipoteichoic acid (LTA), collagen binding protein (ace), hemolysin activator, protein surface, and gelatinases. These factors facilitate their adherence and invasion, thereby triggering or exacerbating inflammatory responses [3]. Gram-negative bacteria, on the other hand, contribute lipopolysaccharides (LPS), bacterial endotoxins crucial in endodontic infections due to their biological effects [3]. LPS induces an immune response in the host, resulting in clinical symptomatology, inflammatory reaction, and the resorption of mineralized tissues. Both LPS and LTA, found in the walls of Gram-negative and Gram-positive bacteria, respectively, possess the ability to strong activate immune cells such as monocytes/macrophages [4,5]. This activation leads to the release of inflammatory cytokines that act on periradicular tissues, causing tissue destruction. The role of LPS and LTA in injuries to dental pulp and periapical tissues has been demonstrated [4,5] in animal models such as rats, as well as using LPS from several clinical strains of *E. faecalis*.

The host mounts an immune-inflammatory response in response to root canal infection, leading to the destruction of the periodontal ligament, cementum, and alveolar bone, with the activation of the osteoclasts [6]. This immune response involves various cells of the immune system, including neutrophils, macrophages, and lymphocytes, which release biochemical mediators [6].

Macrophages, as innate immune cells, possess a dual role, serving as osteoclast precursors and modifiers of the immune response [6]. This dual functionality is attributed, in part, to the release of pro-inflammatory cytokines such as IL-1β, IL-6, and TNF-α. These cytokines play a crucial role in promoting osteoclast formation and activity, leading to periradicular bone resorption. The fusion of monocytes and macrophages ultimately gives rise to osteoclasts, specialized cells highly adapted to bone absorption [7]. Osteoclasts, characterized by their large, mobile, and multinucleated nature, play a pivotal role in the degradation, reabsorption, and remodeling of the bone. The differentiation and activation of osteoclasts are intricately regulated by the balance between the activating receptor of the nuclear factor κβ (RANK), its ligand (RANKL), and their decoy receptor, osteoprotegerin (OPG) [8].

RANK, its ligand (RANKL), and osteoprotegerin play crucial roles in regulating the differentiation, activation, and survival of osteoclasts in both physiological and pathological processes. RANKL induces bone destruction, and its natural decoy receptor, OPG, acts as a protective factor against bone destruction by preventing the binding of RANKL to its receptor RANK. Both RANK and RANKL are expressed in immune system cells, including B cells, activated T lymphocytes, and dendritic cells (DCs) [9]. Osteoprotegerin (OPG), on the other hand, is expressed by osteoblasts as well as immune system cells, such as B cells and dendritic cells. OPG functions to inhibit osteoclastogenesis and the activation of osteoclasts, thereby maintaining the delicate balance of bone turnover [9].

Interleukin 1 receptor-associated kinases (IRAK-1, IRAK-2, IRAK-3, IRAK-4) play a crucial role in coordinating multiple inflammatory pathways that are involved in cell survival, cytokine production, and the priming of the adaptive immune system [10]. Recent studies have highlighted the significance of IRAK signaling pathways in the pathobiology of bone diseases. Consequently, these observations have sparked interest in IRAK kinases as therapeutic target strategies [10,11,12].

In physiological conditions, the bone undergoes continuous renewal through the intricate process of remodeling, wherein osteoclasts contribute to bone resorption and are subsequently replaced by new bone formation by osteoblasts. However, in pathologic situations, this delicate balance is disrupted, leading to an elevated proportion of osteoclastogenesis compared with osteoblastogenesis [13]. A notable example of such dysregulation occurs in apical periodontitis. In chronic apical periodontitis, the prominent feature is the resorption of the alveolar bone in the apical region, a phenomenon driven by the involvement of multiple inflammatory mediators [13].

The objective of this study was to explore the impact of an IRAK-4 inhibitor on the RANK/RANKL pathways in the macrophages THP-1 and peripheral blood mononuclear cells (PBMC) obtained from patients with apical periodontitis when exposed to *Enterococcus faecalis* and its major virulence factor, the lipoteichoic acid (LTA).

## 2. Results

### 2.1. Patients

Clinical and demographic data from patients with apical periodontitis were included in this study are shown in Table 1.

### 2.2. Expression of RANK+ Cells of THP-1 Macrophages

Thp-1 macrophages from the cellular culture, under various culture conditions, were subjected to an analysis for the expression of RANK+ cells by flow cytometry, as outlined in the Section 4. Notably, a significant increase in the percentage of RANK+ cells was observed in the culture in Thp-1 macrophages when exposed to *Enterococcus faecalis*, as compared with all other culture conditions (LTA, α-IRAK-4, α-IRAK-4 + E.f., α-IRAK-4 + LTA) with median and interquartile ranges values of 22.25% (20.10–33.7%), 5.2% (3.42–6.07%), 7.5% (5.87–8.9%), 2.96% (2.08–4.82%), and 3.67% (3.09–8.35%), respectively (Figure 1B, *p* < 0.05).

### 2.3. Expression of RANK+, RANKL+, OPG+ Cells of PBMC from Patients with Apical Periodontitis

PBMC from patients with apical periodontitis, subjected to various culture conditions, were analyzed for the expression of RANK+ cells. Notably, a significant difference in the percentage of RANK+ cells was observed in PBMC when exposed to *E. faecalis*, as compared with all other culture conditions (LTA, α-IRAK-4, α-IRAK-4 + E.f., α-IRAK-4 + LTA). The median and interquartile ranges values for these conditions were 29.35% (24.8–44.13%), 9.4% (8.63–12.45%), 13.6% (10.6–18.5%), 3.38% (2.71–4.24%), and 5.99% (5.56–7.39%), respectively (Figure 2A, *p* < 0.05). We observed an increased percentage of OPG+ cells in PBMC when LTA and α-IRAK-4 were added to the culture, in comparison with control conditions (7.71%, 7.0–12.65%; 7.15%, 5.22–9.29%; 2.05%, 1.28–2.68%, respectively, median and interquartile range), as depicted in Figure 2B (*p* < 0.05). Furthermore, when analyzing double-positive cells for RANK and RANKL, a significant increase was noted in all tested conditions (*Enterococcus faecalis*, LTA, α-IRAK-4 + E.f., α-IRAK-4 + LTA) compared with controls, except for the condition where α-IRAK-4 was added (33.9%, 23.43–37.93%; 14.95%, 11.25–23.25%; 8.0%, 7.33–9.69%; 11.50%, 7.18–13.85%; 4.05%, 2.84–5.86%, median and interquartile range, respectively), as shown in Figure 2C (*p* < 0.05). In contrast, for double-positive cells RANK and OPG, we observed an increase in the percentage of these cells in conditions where LTA, α-IRAK-4, and α-IRAK-4 + LTA were added, in comparison with controls (12.45%, 8.72–19.48%; 8.8, 6.04–11.75%; 7.06%, 5.92–14.30%, median and interquartile range, respectively), as illustrated in Figure 2C (*p* < 0.05). Additionally, we classified PBMC from patients with apical periodontitis based on oral symptomatology and measured the percent of RANK+ cells without stimulation (un-cultured cells). No significant differences were observed between symptomatic or asymptomatic patients (44%, 38.75–47.55%; 50.32%, 35.45–59.9%, median and interquartile range, respectively), as shown in Figure 3A (*p* < 0.05). When comparing the percent of RANK+ cells in THP-1 vs. PBMC from patients with apical periodontitis, we observed an increase in %RANK+ cells when α-IRAK-4 was added to PBMC in comparison with THP-1 when α-IRAK-4 + E.f. and α-IRAK-4 + LTA were added (13.6%, 10.6–18.5%; 2.96%, 2.08–4.82%; 3.67%, 3.09–8.35%, median and interquartile range, respectively), as depicted in Figure 4A (*p* < 0.05). Moreover, the percentage of RANK+ cells was compared between PBMC from patients in culture with *E. faecalis* vs. fresh PBMC (un-cultured PBMCs) from patients. We observed a significantly higher percentage of RANK+ cells in un-cultured PBMCs compared with cultured PBMCs (42.35 ± 9.14 vs. 32.1 ± 13.22, mean ± SD, respectively, *p* = 0.045) in Figure 4B.

### 2.4. Quantification of Inflammatory Cytokines

The concentrations of the inflammatory next inflammatory cytokines IL1β, IL-6, IL-8, IL-12p70, TNF-α, and IL-10 were determined. Notably, an increase in the concentration of IL-8 was observed in Thp-1 cells when α-IRAK-4 + *E. faecalis* was added to the culture, compared with the condition where only α-IRAK-4 was added (5771 ± 614 μg/mL vs. 2522 ± 1541 μg/mL, mean ± SD, respectively, *p* = 0.0061), as illustrated in Figure 5A. Similar results were noted in PBMCs, where the concentration of IL-8 increased in conditions where α-IRAK-4 + *E. faecalis* and *E. faecalis* were added, compared with the condition with α-IRAK-4 + LTA (5082 ± 1131 μg/mL, 4680 ± 853.7 μg/mL vs. 930.9 ± 574 μg/mL, mean ± SD respectively, *p* = 0.0058), as depicted in Figure 5B. However, the concentrations of the other tested cytokines (IL1β, IL-6, IL-12p70, TNF-α, and IL-10) did not show significant differences. 

## 3. Discussion

Several studies have explored the impact of *Enterococcus faecalis* on bone loss and inflammation during the pathogenesis of apical periodontitis pathogenesis [14]; however, the mechanism involved in these processes remains incompletely understood. Therefore, our research aimed to provide a deeper understanding of the intricate interplay between microbial exposure and the RANK/RANKL/OPG pathway in Thp-1 macrophages and PBMCs from patients with endodontic infections. We specially investigated the inhibitory effects of α-IRAK-4 to shed light on previously unknown aspects of oral pathology. This study not only contributes to a more comprehensive understanding of the pathophysiology but also identifies potential treatment targets for managing periapical lesions associated with endodontic infections.

We observed elevated levels of RANKL in Thp-1 cells upon stimulation with *E. faecalis.* While further research is necessary to ascertain the specific implications of these abnormalities in the etiology of apical periodontitis, we believe that they contribute to the initiation and progression of this condition. Notably, our previous research demonstrated that Thp-1 cells stimulated with *E. faecalis* induce osteoclast differentiation and increase the production of pro-inflammatory cytokines such as IL-6 and TNF-α [14]. Similarly, another study revealed that RAW264.7 cells stimulated with *E. faecalis* undergo osteoclastogenesis through a RANKL-dependent pathway [15]. Numerous studies have linked immune system cells, such as macrophages and T cells, to osteoclast function, suggesting that osteoclast precursors regulate cytokine production and bone resorption [16,17]. In our investigation, we found that PBMCs from patients, when stimulated with *E. faecalis,* exhibited increased levels of RANK/RANKL and decreased levels of RANK/OPG-positive cells, indicating the activation of the bone resorption pathway. However, we did not observe any change in RANK expression between symptomatic or asymptomatic patients, suggesting that peripheral macrophages may not be skewed toward osteoclastogenesis. Therefore, it would be very interesting to evaluate symptomatic and asymptomatic patients in the future, since it would be expected that those patients who present symptoms of pain, percussion, etc., would show some changes in these defense mechanisms. However, new studies should be addressed in this context.

To date, there is limited information regarding the inflammatory reactions of macrophages stimulated with *E. faecalis* [18]. However, the impact of *E. faecalis* exposure on osteoclastogenesis remains controversial in the existing literature [15,19,20,21]. Further investigations to elucidate immune interactions within PBMCs and the dual role of macrophages in this context are crucial for a deeper understanding of these complex processes.

Lipoteichoic acid (LTA) stands out as a pivotal factor of *E. faecalis*, playing a major role in refractory apical periodontitis and triggering inflammatory responses both in vivo and in vitro [21,22]. Our findings indicate that Thp-1 and PBMCs stimulated with LTA exhibited decreased levels of RANK/RANKL, concomitant with an increased RANK/OPG ratio. Existing evidence suggests that LTA from *E. faecalis* may inhibit macrophage differentiation into an osteoclast, offering a plausible explanation for our observed results, which lean towards inhibition rather than activation in pathways related to osteoclastogenesis [20,21,22,23]. The role of LTA in osteoclastogenesis remains poorly understood and is still controversial, reflecting the need for further exploration.

It is worth noting that LTA can be classified into various types based on repeated D-alanine contents, and the acyl chain in LTA varies among bacteria species. For instance, LTA from *E. faecalis*, *S. aureus*, and *S. pneumoniae* induces inflammatory responses through the production of TNF-α, nitric oxide, and IL-6. Conversely, LTA from other bacteria, such as *Lactobacilli* spp., exhibits anti-inflammatory properties [5,20]. Additionally, LTA from pneumococcal and staphylococcal origins has different immunogenic capacities in stimulating TLR2 [24]. Therefore, the structural variations in LTA may be a critical factor in determining the pathogenicity of each strain and species.

To comprehensively understand the impact of *E. faecalis*-derived LTA on the dental bone resorption process and inflammation [25], this area of research holds the promise of unraveling the intricate dynamics between bacterial virulence factors and host response in the context of apical periodontitis.

IRAK-4 is a serine–threonine kinase considered essential for signaling in the MyD88-depending pathway of the Toll-like/IL-1-receptor (TIR), as well as in another Toll-like receptor (TLR), such as TLR-2. Upon interaction of the *IRAK-4* gene with *MyD88*, it triggers the activation of the nuclear factor-κb (NF-κB) pathway, subsequently leading to the transcription of inflammatory mediators, including IL-6 [10]. The incorporation of IRAK-4 inhibition to assess its role in the Toll-like receptor signaling pathway is crucial for evaluating the activation of inflammatory and resorption bone responses, including the RANK/RANKL pathway. Our findings revealed decreased levels of RANK-positive cells when stimulated with either *E. faecalis* or LTA. These data suggest, for the first time to our knowledge, that IRAK-4 suppresses the macrophage polarization into an osteoclast by reducing RANKL and increasing OPG. In support of our observations, a recent study demonstrated that IRAK-4 inhibition serves as a potent immunomodulator in the monocyte–macrophage polarization into an osteoclast, particularly in the context of peri-implant osseointegration [26,27].

Moreover, our investigation indicates that Thp-1 and macrophages from PBMCs of patients with apical periodontitis respond differently to the IRAK-4 inhibitor. Therefore, it might be in future studies to delve into the use of IRAK-4 inhibitors on macrophages isolates from gingival tissue and/or damaged tissue resulting from apical periodontitis. This approach could provide further insights into the specific interactions and responses within the local tissue microenvironment, contributing to a more nuanced understanding of the role of IRAK-4 in the context of apical periodontitis.

In response to infections, cytokines are released to control and maintain homeostasis. At local areas of inflammation, IL-8 is a pro-inflammatory cytokine that is produced by various cells, primarily by immune innate cells. It is rapidly synthesized to recruit, activate, and retain inflammatory cells, including monocytes and macrophages. This study revealed increased levels of IL-8 in the supernatants of THP-1 macrophages and PBMCs stimulated with *E. faecalis* and the α-IRAK-4 inhibitor while lowering the expression in LTA streptococcus derived in combination with the IRAK-4 inhibitor. Conversely, a lower expression of IL-8 was observed in the presence of LTA derived from streptococcus, particularly in combination with IRAK-4 inhibitors.

The heightened IL-8 levels observed may be associated with the distinct virulence factors of *E. faecalis* rather than solely with LTA exposure. Interestingly, we found that LTA reduced IL-8 levels upon the addition of the IRAK-4 inhibitor, suggesting the IRAK-4-dependent pathway of IL-8 reduction. Previous studies have described that *Staphylococcus aureus*-derived LTA stimulates IL-8 release through CD14-dependent pathways rather than TLR2- and TLR4-mediated pathways [28,29,30]; similar results have been observed in other bacteria such as Streptococcus and Actinomyces, which are implicated to in endodontic infections and apical periodontitis [29]. Additionally, the nature of LTA has been shown to drive different immune responses, emphasizing the importance of exploring this aspect in future investigations.

While this study offers valuable insights, due to the importance of studying microbiological and immunological factors involved in oral infections with bone loss, which are common in our population, it is important to acknowledge several limitations that warrant consideration in future research. Expanding the sample size and including patients with various endodontic pathologies would enhance the generalizability of our findings. Additionality, investigating the immunogenicity of antigens involved in inflammatory response and the osteoclastogenic process in periodontal disease will be crucial for advancing future investigations. Exploring additional markers or factors associated with symptomatology could provide a more comprehensive understanding of the complex interactions in the context of endodontic infections.

## 4. Materials and Methods

Patients. Twenty patients with apical periodontitis, either symptomatic or asymptomatic, without systemic diseases, were recruited from the Endodontic Clinic of the Endodontics Postgraduate Program, Faculty of Dentistry, Autonomous University of San Luis Potosi, in 1 year. Diagnosis was established by endodontists through a comprehensive set of examinations, including thermic tests (cold response), percussion, palpation, mastication assessment, and X-ray analysis, to confirm the presence of an evident periapical lesion, where X-ray shows radiolucence at the apex zone and a widening of the periodontal ligament. Peripheral blood samples (6 mL) were collected in ethylene-diamine-tetra acetic acid (EDTA) tubes for the isolation of mononuclear cells from patients. This isolation was achieved using a density gradient with Ficoll-Paque (GE Healthcare, Uppsala, Sweden). Cellular viability was determined by microscopic examination using trypan blue staining. All patients provided detailed histories and signed an informed consent form. Comprehensive patient data, including clinical history and diagnostic data of patients, are shown in Table 1. The approval for this study was obtained by the Faculty of Dentistry’s Ethical and Investigation Committee, which approved this study (CEIFE-019-018).

Cellular line and differentiation. The THP-1 cell line, a human monocyte/macrophage leukemia (Sigma-Aldrich, Darmstadt, Germany), was utilized for cellular culture. A total of 5.0 × 10^5^ cells were seeded in 24-well plates with a flat bottom, using RPMI-1640 culture medium supplemented with 10% FBS, 1% glutamine, and antibiotics (streptomycin/penicillin). The Thp-1 cells were incubated at 37 °C, 5.0% CO_2_, and 100% humidity for 5 days. Phorbol myristate acetate (PMA) was added at Thp-1 cells at a concentration of 50 ng/mL on days 1–3 to induce differentiation into activated macrophages. Cell viability was assessed by microscopy using 1% trypan blue staining for the same evaluator, which consistently achieved viability > 80%. Once this procedure was carried out, the Thp-1 cells were given the ability to phagocytose bacteria, using the CFSE assay, explained below.

Phagocytosis assays. Phagocytosis assays were conducted to assess the internalization at THP-1 cells of carboxy-fluorescein (CFSE)-labeled bacteria (Thermo Fisher Scientific, Waltham, MA, USA) at a concentration of 5.0 µM of CFSE, performed over 20 min at 4 °C. Different amounts of labeled *Enterococcus faecalis* bacteria (at the 0.5 McFarland scale with multiplicity of infection (MOI), 10, 100, and 1000 were mixed with different numbers of THP-1 macrophages (10 × 10^5^, 5.0 × 10^5^, and 2.5 × 10^5^ cells) and incubated at 37 °C with 5% CO_2_ for 2 h. Subsequently, the Thp-1 macrophages were washed, centrifuged at 1500 rpm for 5 min, and analyzed using a FACSCanto II flow cytometer (BD, San Diego, CA, USA), using the FlowJo software v10.0 (BD, San Diego, CA, USA). Cellular viability was assessed before and after the phagocytosis using trypan blue staining, and microscopy was employed for this purpose.

Cellular cultures. In a 24-well plate, Thp-1 macrophages were added at a multiplicity of infection (MOI) of 10, with a concentration 5 × 10^5^ cells, under different culture conditions: only macrophages (serving as controls), macrophages with Staphylococcal lipoteichoic acid (LTA) at a concentration 5 µg/mL (Sigma, Life Sciences, Darmstadt, Germany), macrophages with *E. faecalis* (isolated from root canals of patients with secondary endodontic infections), macrophages treated with a monoclonal antibody inhibitor of α-IRAK-4 (50 mM) (Santa Cruz Biotechnology, Santa Cruz, CA, USA), macrophages treated with α-IRAK-4 + LTA, and macrophages treated with α-IRAK-4 + *E. faecalis*. For conditions involving *E. faecalis*, the microorganisms were added for 2 h, after which the wells were thoroughly washed to eliminate any residual bacteria. The plate was then incubated at 37 °C with 5% CO_2_ for a total period of 48-h. In addition to Thp-1 macrophages, the same culture conditions were replicated using PMBCs from patients with apical periodontitis.

Flow cytometry. Following the 48-h culture period, cells (Thp-1 and PBMCs from patients, both) and supernatants were collected for analysis. The cells were detached using a versene solution for 10 min, followed by washing and placement in cytometer tubes. Simultaneously, the supernatants were carefully collected and stored at −80 °C for the subsequent analysis of inflammatory cytokines. For cellular analysis, the collected cells were treated with 1% paraformaldehyde and then stained with α-RANK/PE (BioLegend, San Diego, CA, USA), α-RANKL/FITC (BioLegend), and α-OPG-Alexa 555 (BioLegend) for 30 min at 4 °C in darkness. Subsequently, the stained cells were processed for flow cytometry using a FACSCanto II flow cytometer, acquiring a minimum of 100,000 events. The acquired data were then analyzed using the FlowJo software v10.0 (San Diego, CA, USA).

Quantification of inflammatory cytokines. Cell-free supernatants from Thp-1 macrophages were collected and promptly stored at −80 °C for subsequent cytokine quantification. The levels of specific cytokines including IL-1β, TNF-α, IL-6, IL-8, IL-10, and IL-12p70 were determined by using a cytometric bead array kit (CBA) from BD, Biosciences (San Diego, CA, USA). The quantification was performed according to the manufacturing instructions. For cytokine analysis, flow cytometry analysis was employed with an Accuri C6 cytometer, and the obtained data were analyzed with FCAP Array software v3.0 (BD, Biosciences, San Diego, CA, USA).

Statistical analysis. The sample size for the patients was calculated by using the software N v5.0 (GraphPad, San Diego, CA, USA), with 80% of the power of study. The data are presented as either the arithmetic mean and standard deviation (SD) or the median and interquartile range (Q1–Q3), depending on the normality of the distribution. To assess differences between the two groups, Student’s t-test or the Mann–Whitney U test was employed. For comparisons among three or more groups, one-way ANOVA or the Kruskal–Wallis sum-rank test was used based on the normal or no-normal distribution, respectively. Post hoc analyses included Tukey’s multiple comparison test and/or Dunn´s multiple comparison test. The statistical analysis was conducted using the GraphPad Prism v5.0 software (GraphPad, San Diego, CA, USA). A significance level of *p* < 0.05 was considered statistically significant.

## 5. Conclusions

In this study, we observed that the IRAK-4 inhibitor decreases the RANK/RANKL/OPG pathway, possibly by inhibiting the LTA-TLR2 signaling pathway and increasing OPG expression. This modulation could potentially halt osteoclastogenesis in both apical periodontitis patients and Thp-1 cells. Our findings contribute to a better understanding of the bone resorption process and suggest that inhibiting IRAK-4 could be a therapeutic strategy to combat oral infections caused by *E. faecalis* in apical periodontitis or other oral diseases characterized by bone destruction. However, further studies are needed to investigate whether targeting IRAK-4 provides an effective approach for treating immune-inflammatory responses and bone resorption in oral diseases in vivo.

## Figures and Tables

**Figure 1 ijms-25-08434-f001:**
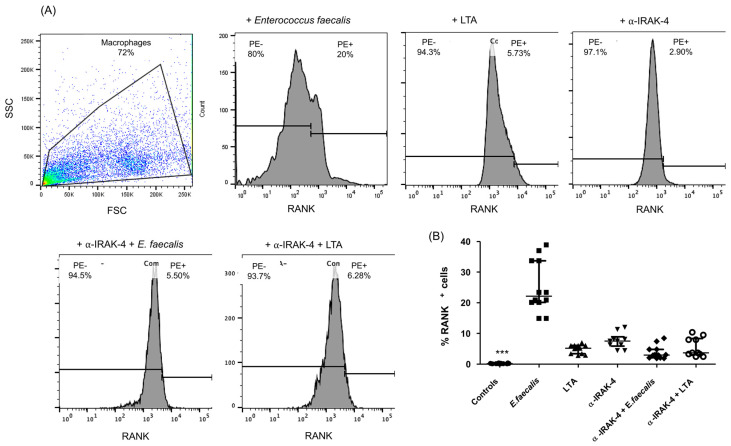
Analysis of the percentage of RANK+ cells in Thp-1 macrophages. The macrophages were exposed to several stimuli, as indicated in the Section 4, and the levels of RANK+ cells were measured by flow cytometry. (**A**) Flow cytometry image of a representative experiment. The percentages of RANK+ cells are indicated. (**B**) Percentage of RANK+ cells in cultured macrophages with bacteria, LTA, and the inhibitor IRAK-4. Data correspond to the median and interquartile range, *n* = 20. *** *p* < 0.001. SSC = Side scatter, FSC = Forward scatter.

**Figure 2 ijms-25-08434-f002:**
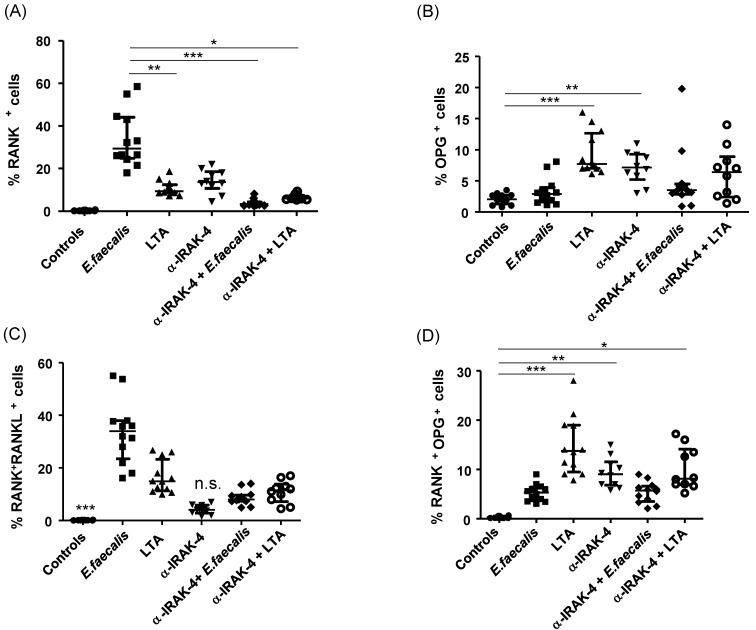
Analysis of the percentages of RANK+, OPG+, and RANKL+ cells in PBMCs exposed to LTA, bacteria, and α-IRAK-4 from patients with apical periodontitis. The PBMCs from patients were exposed to several stimuli, as indicated in the Section 4, and the levels of RANK+, OPG+, and RANKL+ cells were measured by flow cytometry. (**A**) Percentage of RANK+ cells in cultured macrophages with bacteria, LTA, inhibitor IRAK-4. (**B**) Percentage of OPG+ cells in cultured macrophages with bacteria, LTA, and inhibitor IRAK-4. (**C**) Percentage of RANK+RANKL+ cells in cultured macrophages with bacteria, LTA, inhibitor IRAK-4. (**D**) Percentage of RANK+OPG+ cells in cultured macrophages with bacteria, LTA, and inhibitor IRAK-4. Data correspond to the median and interquartile range, *n* = 20. * *p* < 0.05, ** *p* < 0.01, *** *p* < 0.001.

**Figure 3 ijms-25-08434-f003:**
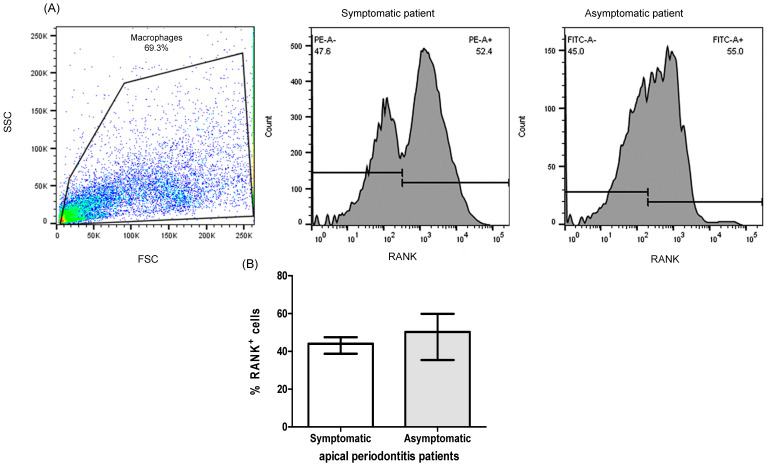
Analysis of the percent of RANK+ cells in PBMCs from patients with apical periodontitis. The isolated PBMCs from patients were stained for RANK, and the quantification was performed by flow cytometry. (**A**) Flow cytometry image of a representative experiment. The percentages of RANK+ cells are indicated. (**B**) Percentage of RANK+ cells in PBMCs from patients with apical periodontitis classified as symptomatic and asymptomatic. Data correspond to the median and interquartile range, *n* = 20.

**Figure 4 ijms-25-08434-f004:**
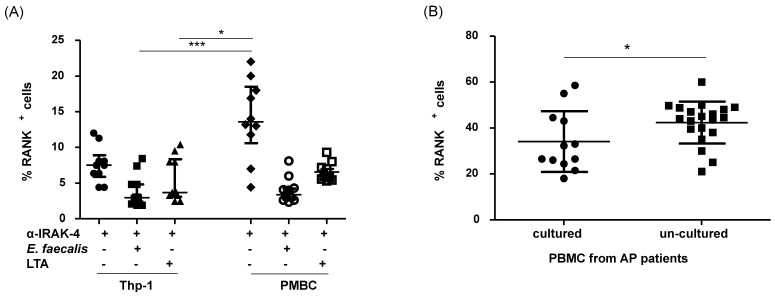
Comparison between the percentages of RANK+ cells in PBMCs in culture vs. un-cultured PBMCs from patients with apical periodontitis. (**A**) Percentage of RANK+ cells in PBMCs from patients with apical periodontitis. (**B**) Percentage of RANK+ cells in PBMCs from patients with apical periodontitis, freshly isolated (un-cultured) and cultured PBMCs. Data correspond to the median and interquartile range, *n* = 20. * *p* < 0.05, *** *p* < 0.001.

**Figure 5 ijms-25-08434-f005:**
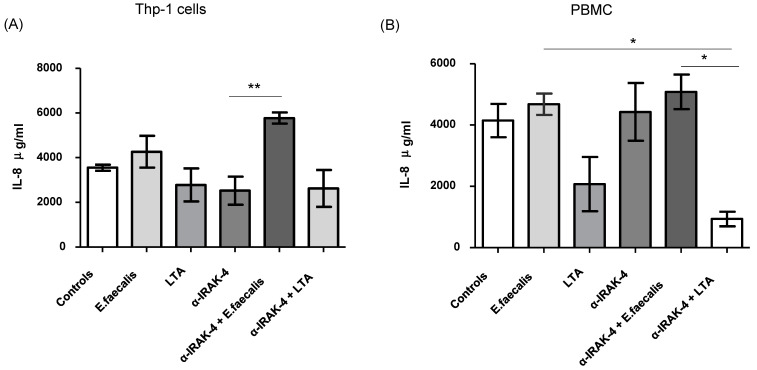
Quantification of inflammatory cytokines. Supernatants of THP-1 macrophage cultures were obtained and assayed for the presence of the indicated cytokines by flow cytometry analysis, as indicated in the Section 4. (**A**) Concentration of IL-8 in ug/mL in Thp1 cells with the different culture conditions. (**B**) Concentration of IL-8 in ug/mL in PBMC from patients with AP with different culture conditions. Data correspond to the mean ± standard deviation, *n* = 20. * *p* < 0.05, ** *p* < 0.01.

**Table 1 ijms-25-08434-t001:** Clinical and demographic data of patients with apical periodontitis.

Demographic or Clinic Characteristics	Patients *n* = 20
Gender (%)	
**Female**	85%
**Male**	15%
**Age (years)**	45.0 ± 8.0 years
**Patients with periradicular lesion (%)**	100%
**Apical periodontitis (%)**	
**Primary infection**	40%
**Secondary/persistent infection**	60%
Symptomatic patients (%)	83%
Palpation 100%	
Percussion 100%	
Mastication 100%	
Mobility 100%	
Asymptomatic patients	17%
**Evolution time of endodontic treatment (years)**	4.5 ± 4.7 years

## Data Availability

All data generated or analysed during this study are included in this published article.

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
