# Peer review of "α-IRAK-4 Suppresses the Activation of RANK/RANKL Pathway on Macrophages Exposed to Endodontic Microorganisms"

_ijms, 2024, doi:10.3390/ijms25158434_

Round 1

Reviewer 1 Report

Comments and Suggestions for Authors

Dear Authors,

Congratulations on the job you have done and presented in this manuscript. I believe that your work can be of high interest for the general reader. However, there are major revisions required before consideration for publication in a such high quality journal. Please see the attachment

Reviewer 2 Report

Comments and Suggestions for Authors

Dear authors,
thank you for the opportunity to revise this manuscript. The topic is of great interest.

Please revise the following aspects.

Abstract

Please divide in subsections and reduce background. Please better explain M&M.

Introduction

-“The 67 role of LPS and LTA in injuries to dental pulp and periapical tissues has been demontrated”: please add some content to this sentence.

-Please move content about mechanisms of IRAK-1, IRAK-2, IRAK-3, IRAK-4 to Discussion section.

M&M

-Please add number of Ethical committee

Results

Please give more resolution to each part of Figure 1 and 3.

Discussion

-“However, we did not observe any change in RANK expression between symptomatic or asymptomatic patients,  suggesting that peripheral macrophages may not be skewed toward osteoclastogenesis“: please add content to this observation between symptomatic or asymptomatic patients.

-“Previous studies have described that Staphylococcus aureus-derived LTA stimulates IL-8 release through CD14-dependent pathways rather than TLR2 and TLR4-mediated pathways”: please add comparison between E. faecalis and other possible retrieved bacteria.

Comments on the Quality of English Language

Moderate English checking is required

Reviewer 3 Report

Comments and Suggestions for Authors

Some improvements that could be made to the manuscript:

The authors must specify in Mat % Met what criteria were followed for the diagnosis of periapical pathology. Was a radiographic examination performed? What radiographic criteria were followed for the diagnosis of apical periodontitis?

Was the evaluation of cell viability with trypan blue staining and microscopy blinded?

Round 2

Reviewer 1 Report

Comments and Suggestions for Authors

Dear Authors, 

I believe that you took advantage on the reviewers comments and improved the manuscript accordingly. I have no further comments.